# The Gut Microbiome as a Key Determinant of the Heritability of Body Mass Index

**DOI:** 10.3390/nu17101713

**Published:** 2025-05-18

**Authors:** Thomas M. Barber, Stefan Kabisch, Andreas F. H. Pfeiffer, Martin O. Weickert

**Affiliations:** 1Warwickshire Institute for the Study of Diabetes, Endocrinology and Metabolism, University Hospitals Coventry and Warwickshire, Clifford Bridge Road, Coventry CV2 2DX, UK; t.barber@warwick.ac.uk; 2Division of Biomedical Sciences, Warwick Medical School, University of Warwick, Coventry CV4 7AL, UK; 3Human Metabolism Research Unit, University Hospitals Coventry and Warwickshire, Clifford Bridge Road, Coventry CV2 2DX, UK; 4Department of Endocrinology and Metabolic Medicine, Campus Benjamin Franklin, Charité University Medicine, Hindenburgdamm 30, 12203 Berlin, Germanyafhp@charite.de (A.F.H.P.); 5Deutsches Zentrum für Diabetesforschung e.V., Geschäftsstelle am Helmholtz-Zentrum München, Ingolstädter Landstraße, 85764 Neuherberg, Germany; 6Centre for Sport, Exercise and Life Sciences, Faculty of Health & Life Sciences, Coventry University, Coventry CV1 5FB, UK

**Keywords:** gut microbiota, BMI, heritability, appetite, metabolism

## Abstract

The pathogenesis of obesity is complex and incompletely understood, with an underlying interplay between our genetic architecture and obesogenic environment. The public understanding of the development of obesity is shrouded in myths with widespread societal misconceptions. Body Mass Index (BMI) is a highly heritable trait. However, despite reports from recent genome-wide association studies, only a small proportion of the overall heritability of BMI is known to be lurking within the human genome. Other non-genetic heritable traits may contribute to BMI. The gut microbiome is an excellent candidate, implicating complex interlinks with hypothalamic control of appetite and metabolism via entero-endocrine, autonomic, and neuro-humeral pathways. The neonatal gut microbiome derived from the mother via transgenerational transmission (vaginal delivery and breastfeeding) tends to have a permanence within the gut. Conversely, non-maternally derived gut microbiota manifest mutability that responds to changes in lifestyle and diet. We should all strive to optimize our lifestyles and ensure a diet that is replete with varied and unprocessed plant-based foods to establish and nurture a healthy gut microbiome. Women of reproductive age should optimize their gut microbiome, particularly pre-conception, ante- and postnatally to enable the establishment of a healthy neonatal gut microbiome in their offspring. Finally, we should redouble our efforts to educate the populace on the pathogenesis of obesity, and the role of heritable (but modifiable) factors such as the gut microbiome. Such renewed understanding and insights would help to promote the widespread adoption of healthy lifestyles and diets, and facilitate a transition from our current dispassionate and stigmatized societal approach towards people living with obesity towards one that is epitomized by understanding, support, and compassion.

## 1. Introduction

The effective prevention and management of any disease usually require deep knowledge and insight into its underlying aetiopathogenesis. This includes the design of effective preventive and therapeutic strategies that target key pathways within this pathogenic arena, as illustrated by ‘cardiovascular-kidney-metabolic syndrome’ [1]. In the case of monogenic conditions, the underlying aetiopathogenesis is often clearly understood, serving as a beacon for potential therapeutic strategies. Furthermore, with the emergence of genetically targeted therapies delivered by adenoviruses and the like [2], it is likely that in the future, monogenic conditions as a group will be amongst the most treatable of diseases, even curable in some cases. At the other end of the pathogenic spectrum sit the many chronic diseases that typify our modern-day era, which, far from originating from a single gene mutation, have complex and multifaceted causal pathways, implicating interplay between genetic and environmental components, often activation of the inflammasome [3], and which in many cases are incompletely understood. Obesity is one such disease. Unfortunately, the complex and opaque aetiopathogenesis of conditions like obesity poses unique challenges in the search for effective preventive and management strategies.

Obesity is the most important health crisis of our time and is classified as a non-infectious global pandemic by the World Health Organisation (WHO). The global prevalence of obesity exceeds 890 million, and overweight affects 2.5 billion adults [4]. The dysmetabolic sequelae of obesity stem from its association with insulin resistance (IR), particularly in the context of visceral adiposity and ectopic fat deposition [5]. There are >50 obesity-related conditions that include, most notably, type 2 diabetes mellitus (T2D), and others like hypertension, dyslipidaemia, polycystic ovary syndrome (PCOS) [6], obstructive sleep apnoea (OSA), and metabolic-associated fatty liver disease (MAFLD) [7]. Furthermore, obesity is an important risk factor for many malignancies [8,9], and confers a substantial health economic burden on humanity, stemming from direct and indirect costs [10]. Obesity impairs work productivity [11], psychosocial functioning [12], and overall wellbeing and quality of life.

An unfortunate consequence of any common modern-day chronic disease with a complex and incompletely understood underlying pathogenesis is that this provides an ideal opportunity for the generation and propagation of myths. This is perhaps illustrated by obesity more so than any other chronic disease [13]. The media portray obesity as resulting from our lifestyle choices—it is assumed that we all have the choice of what and when to eat, and how much to eat, and it is the unique will of each of us to control our own appetite [14]. The corollary of this oversimplistic and frankly false message is that weight gain therefore results from an inability to control one’s own appetite through vices like greed and gluttony, and this is compounded perhaps by laziness and an unwillingness to engage in physical exercise and activity. This is a common myth that is widespread within our society and is even shared by some healthcare professionals [15]. In the context of a complex and incompletely understood pathogenesis, it is easy to understand the popularisation of such a myth. Furthermore, as with many conspiracy theories in our modern-day world, it is easy to see the attraction of such a myth, providing a simple explanation for weight gain and obesity that everyone can understand and relate to, that implicitly and conveniently apportions blame to the individual living with obesity, and which is intellectually lazy and requires little effort to understand. As humans, we are generally averse to uncertainty and unknowns, and we seem much happier living with a simple explanation for phenomena that are incorrectly perceived as simple [16], such as obesity, even if this explanation is clearly false. (Human history is littered with such examples!)

It is our view that the widely held misconception regarding the pathogenesis of weight gain and obesity outlined here, with its inherent blame and focus on human vices, has contributed towards a widespread dispassionate perspective on obesity and its associated stigma [17]. Furthermore, this obesity myth also likely contributes to a relative lack of funding for effective therapies for obesity. This is particularly evident in recent times, with a gaping disconnect between the unprecedented interest and development of novel incretin-based pharmacotherapies for obesity (with ‘Big Pharma’ and >200 biotechnology industries globally interested in the obesity space) and the relative lack of funding and reimbursement to support the widespread administration and adoption of such therapies at a community-focused level (both within NHS and healthcare settings globally). Whilst focus on treatment strategies for obesity is important, we argue that the development of a clear understanding of and insight into the pathogenesis of obesity is also important to guide future preventive and therapeutic strategies, but also to help address the widespread misconceptions of obesity within our society, reduce and eliminate stigma, and ultimately foster a more compassionate approach towards people living with obesity. Moreover, despite the substantial advances in the weight-loss efficacy of pharmacotherapies (particularly those based on glucagon-like peptide-1 (GLP1) and other incretin and intestinal hormones), these therapies have limitations, including gastrointestinal side effects and weight re-gain following their discontinuation [18]. It is important that we maintain emphasis on lifestyle factors (including nutrition) as preventive and therapeutic strategies for obesity.

Whilst much published work highlights the obesogenic environment [19], in this concise narrative review, we focus on the heritability of Body Mass Index (BMI) to illustrate that obesity has an important genetic component, an insight that is often neglected by and omitted from the popularized myth of obesity pathogenesis outlined above. We discuss the complex interlinks between the gut microbiota and the central hypothalamic control of appetite and metabolism. We look beyond the human genome to explore the hypothesis that the missing heritability of BMI from reported ‘Genome Wide Association Studies’ (GWAS) in obesity stems in part from the heritable gut microbiome and its associated vast metagenome (Figure 1). Finally, we consider the implications of the dynamicity and mutability of the gut microbiome as a promising therapeutic target for the prevention and management of obesity, and even as a possible future maternal pre-conception strategy to help protect offspring from the vicissitude of future weight gain and obesity development.

## 2. Methodology

We performed a narrative review of the current literature, using PubMed for this purpose. The search terms (referring to the title of published articles) were as follows: ‘Body Mass Index; genetics; heritability; gut microbiota; appetite; vaginal delivery; breast feeding’. We only considered articles written in English, and there was no restriction on the date of publication. Due to the extensive number of published articles in this field, it was necessary for us to be selective in the choice of articles for inclusion in this concise narrative review. Where possible, we chose articles published more recently and which have a greater impact within the field.

## 3. The Heritability of BMI

Heritability refers to the quality of a characteristic being transmissible from parent to offspring. Technically, the term ‘heredity’ refers to the specifics of genetic transmission between generations. Conversely, the term ‘heritability’ has a broader meaning, and encompasses the transmission of a biologically interesting phenotype from parent to offspring, and quantifies this level of predictability. Any biological phenotype that has multiple factors contributing to its expression (including genetic and environmental factors) tends to assume a normal distribution, common examples being height and BMI. The heritability of such traits refers numerically to the variation within that normal distribution that is due to biological transmission from parent to offspring. Given the obvious importance of genetics as a transmissible biological trait, and the relatively recent emergence of the field of epigenetics and metagenomics, traditionally, the terms ‘heritability’ and ‘heredity’ have tended to be used synonymously, and heritability to refer only to genetic factors. However, it is important for our discussion within this review to understand and appreciate that heritability can include other biologically transmissible traits that are separate from the human genome.

To gain a deeper understanding of and insight into the pathogenesis of obesity, it is essential to explore the heritability of BMI. The identification of genetic and other biological contributors to weight gain and obesity would provide therapeutic targets for the prevention and management of obesity. Furthermore, such insights would help to address the widely believed mythical explanation for obesity outlined above, in which lifestyle choices and human vices account solely for weight gain and obesity (and independent from any genetic or other biological effects). Any heritability of BMI would argue strongly against such a naïve and simplistic model of weight gain but rather support obesity as a biological disease that is hard-wired in our genetics, and perhaps in our other heritable biological traits. Some of our best evidence for the heritability of BMI stems from twin studies. In one such study from the ‘Chinese National Twin Registry’, 1421 twin pairs were included, and the heritability of BMI was calculated at 72% [20]. Interestingly, in this study, the heritability of both BMI and cardiometabolic traits declined with age, with environmental factors assuming a relatively greater role than genetics in older people [20]. Regarding the phenotypic correlations between BMI and cardiometabolic traits, genetic factors played a consistently greater role than environmental factors [20]. In another population study using data from medical screening of military personnel in Israel, including >447,000 offspring, there was a calculated heritability of 39% between mid-parental and offspring BMI [21]. The clear conclusion from these and other population-wide genetics studies reported in the literature is that BMI, and by implication, obesity, is highly heritable. Overall, the heritability of BMI is estimated to be some 40–50% [22]. However, the heritability of BMI varies according to the BMI subgroup within the population. For normal-weight individuals, the heritability of BMI is around 30% [22]. However, in the subgroup of people with obesity, the heritability of BMI is 60–80% [22]. Furthermore, fat distribution and the presence of ectopic fat are also heritable traits (around 30–55%) [22].

A detailed discussion of the genetic architecture of obesity is beyond the scope of this review and has been covered elsewhere [23,24]. Here, we provide a brief overview. There are two main elements to this discussion. Monogenic forms of obesity are rare and stem, by definition, from a mutation in a single gene that often has major adverse effects on the control of appetite and metabolism and often manifests with substantial weight gain and obesity from an early age. We have identified mutations in at least 15 genes, mainly those that result in deficiencies in the leptin–melanocortin signaling pathway within the hypothalamic appetite centre [22]. The study of the genetics of monogenic obesity has provided much insight into the neurobiological control of appetite and metabolism. In this review, we focus on the second element of obesity genetics: polygenic forms of obesity that are common and underlie the high prevalence of obesity within the populace. Within the realm of polygenic obesity, there is heterogeneity with distinct phenotypic subtypes based on measures of body composition, insulin sensitivity, physical fitness, glycaemia, and cardiovascular risk [25]. However, for this review, we consider polygenic obesity as a single entity.

GWAS studies of obesity have identified >1000 gene variants that have an impact on BMI, with most alleles that influence BMI only contributing a few grams, more or less, to body weight [22,24]. Interestingly, alleles that promote obesity tend to have a greater effect in those individuals with a propensity for weight gain and obesity and exert minimal effects in individuals of normal weight [22]. Therefore, the penetrance of BMI-influencing alleles seems variable according to BMI, although it is not known whether the effect size of BMI-increasing alleles precedes the onset of weight gain and obesity or is enhanced by the obese state [22]. Indeed, gene–environment interaction analyses reveal that our obesogenic environment could be amplifying genetic risk for obesity [23]. Our current genetic model of polygenic obesity promotes the heritability of BMI as being influenced by thousands of gene variants [22], each of which individually has a relatively small effect on body weight, but their overall cumulative effect underlies the phenotypic expression of BMI in each of us. For most GWAS-identified genetic loci that influence BMI, we lack an understanding of causality (translation from variant to function) [24]. However, insights from GWAS reveal that gene variants that impact total body mass are expressed primarily within the central nervous system (CNS), particularly the hypothalamic centres that regulate the control of appetite and metabolism [23]. Conversely, gene variants that impact fat distribution are enriched within the adipose tissue [23].

Data from GWAS have transformed our understanding of polygenic obesity and contributed towards our insight into the central neurobiological control of appetite and metabolism. However, despite the power of reported GWAS on tens of thousands of participants, the gene variants implicated in BMI control only account for a small proportion of the overall heritability of BMI. The ‘polygenic score’ (PGS) is used to assess genetic susceptibility to complex diseases such as obesity. Including data on >2 M gene variants identified from GWAS, the PGS only explains 8.4% of the overall variation in BMI [24]. The low PGS for obesity is redolent of that for other complex diseases based on GWAS data, including, for example, polycystic ovary syndrome (PCOS) [26,27] and T2D [28]. The question then is: where is all the missing heritability of obesity hiding? It is possible that the reported GWAS in obesity simply lack power, and that some of the missing heritability is lurking within many gene variants with very small effect sizes that are only detectable with GWAS of much greater power (that would require hundreds of thousands or even millions of participants to detect). Alternatively, epigenetic effects may contribute to the missing heritability. However, given that heritability refers to the generational transmission of biological traits (not limited to genetic factors), it is important for us to consider non-genetic factors that may mediate part of the missing heritability of BMI. The gut microbiome is an excellent candidate.

## 4. The Gut Microbiome

Human cellular function within well-defined organs and systems forms a firm foundation for our traditional model of human physiology. In recent times, our understanding of physiology has morphed from this traditional human-centric perspective to a vista that accommodates a complex interaction and symbiotic co-evolution over eons of hominid evolution, between human cells and 100 trillion foreign microbes, the latter of which vastly outnumber our own cells. These foreign microbes are referred to collectively as the ‘microbiome’. Although terms like ‘virome’ and ‘mycobiome’ refer to the collection of viruses and fungi, respectively, found in or on an organism, the microbiome is an umbrella term that includes all prokaryotic cells and viruses that associate with an organism, and from our perspective, the human body [29,30]. The microbiota accumulate on any epithelial or endothelial surface that is exposed to the environment, including the genitourinary tract, respiratory epithelia, and skin [31]. However, the vast majority of the human microbiota reside within the gut, and the majority of these (around 70%) exist within the colon [32]. The gut microbiome plays a central role in the regulation of immunoinflammatory pathways, and therefore in the development of many modern-day chronic illnesses. These include chronic inflammatory and auto-immune conditions, atopies, food intolerances, and possibly cardio-metabolic conditions [33], and neuropsychiatric disorders such as Parkinson’s disease [34], autism spectrum disorder [35], chronic pain [36], and disorders of mood and affect [37]. A healthy gut microbiome (eubiosis), outside of the neonatal period, is typified by a diverse and rich array of microbes, achieved and maintained through a high-fibre plant-based diet [32]. Conversely, an unhealthy gut microbiome (dysbiosis) is impoverished and imbalanced, and may also be associated with ‘leakiness’ of the gut epithelium, with the propensity for gut microbes (or components of gut microbes) to translocate across the gut wall into the vascular system (endotoxinaemia) [38]. ‘Leaky gut’ is poorly defined and should not be confused with actual defects in the gut barrier that occur in the context of chronic inflammatory bowel diseases. Furthermore, there is much public misinformation and advertising that promote the use of various prebiotic, postbiotic, and symbiotic supplements to ‘heal leaky gut’ with little supportive evidence. A challenge for the future will be to better define ‘leaky gut’ and to determine its effective and evidence-based management. Related to the gut microbiome, the ‘metagenome’ refers to the collective genome of the human microbiome. Given that the microbiome vastly outnumbers our own cells, it follows that the metagenome is orders of magnitude larger than the human genome (which contains around 20,000 genes) [39].

## 5. Interlinks Between the Gut Microbiome and Central Appetitive and Metabolic Control

As outlined, BMI is heritable. Our genetic architecture is a key mediator of such heritability, particularly regarding the expression of gene variants that influence the central hypothalamic control of appetite and metabolism. In recent times, there has been a renaissance in our understanding of and insight into the control of appetite and metabolism (stemming mainly from rodent-based studies), going beyond human genetics, with the gut microbiome looming as a cynosure [40]. Broadly, the gut microbiota (and/or their metabolic by-products) communicate with the brain both indirectly (via entero-endocrine mechanisms and autonomic afferent pathways within the gut wall itself) and directly via the bloodstream. This latter mechanism requires the translocation of components of the gut microbiota (endotoxins) and/or their metabolic by-products across the gut wall and into the bloodstream, enabling the exertion of both peripheral and central effects. As such, the gut wall forms a boundary between the gut microbiota and human cells [29]. Gut leakiness enables the translocation of microbial wall components, such as lipopolysaccharides (LPS) or microbes, directly. Microbial metabolic components including amino acids, lipid species, and many more may be transported by intestinal transporters. These metabolites probably contribute to beneficial metabolic effects, even though some are unhealthy. To defend and protect the gut wall, there is a layer of mucus (mucin) [29]. In one fascinating human-based study, there was an inverse correlation between the thickness of the colonic mucus (measured as the distance between the gut epithelial lining and the gut microbiota on colonic biopsies) and metabolic measures of BMI and glycaemic indices (HbA1c and fasting glucose levels) [41]. These data are consistent with the gut wall mucus layer having a protective effect on gut wall leakiness and reducing the propensity for weight gain and obesity (and dysmetabolic effects) through optimized central hypothalamic control of appetite and metabolism. In addition to the mechanisms outlined, gut microbiota may also signal to the brain through the release of nanosized extracellular vesicles, via their abundant small RNA and protein cargo, that can have direct effects (via crossing the blood–brain barrier) on the regulation of gene expression within the brain and both neuroinflammatory and neurodegenerative processes [42].

The gut microbiome influences the propensity for weight gain, BMI, and insulin sensitivity via the central control of appetite and metabolism. The composition of the gut microbiome correlates with body weight, with lean and obese people having distinct gut microbial compositional phenotypes [43]. In a systematic review and meta-analysis of >1900 participants, it was demonstrated that weight loss was associated with statistically significant increases in α-diversity of the human gut microbiome and reductions in intestinal permeability [44]. Weight loss also resulted in an increase in the relative abundance of *Akkermansia*, but no changes in individual species, phyla, or faecal short-chain fatty acids (SCFAs) [44]. In addition to changes in body weight, lifestyle and dietary factors, particularly the intake of dietary fibre, also influence gut microbiota composition [45,46]. In a human-based study, a high-fibre (oligofructose) diet reduced the content of Gram-negative gut bacteria and body weight, whereas a high-fat diet increased the content of Gram-negative bacterial LPS within the gut [47]. In the same study, continuous subcutaneous infusion of LPS for 4 weeks increased body weight, hepatic fat content, and markers of both insulin resistance and inflammation, comparable to following a high-fat diet [47]. These data reveal that dietary fibre consumption influences the composition of the gut microbiota (and the relative proportions of gut microbiota species), thereby linking the gut microbiota with insulin sensitivity, inflammation, and the central control of appetite and metabolism [43]. Human-based studies have also shown some favourable effects of faecal transplantation (FT) on glucose tolerance, but disappointing weight loss effects of FT in human recipients with obesity [48,49]. Conversely, rodent-based studies reveal effects of FT on both metabolic status and body weight [48]. In obese rodent models, weight loss resulted in concurrent changes in gut microbiota composition towards a lean phenotype [43,50]. Furthermore, rodent-based studies involving FT from obese or lean mice to germ-free mice reveal that only the recipients of gut microbiota from obese mice exhibited a subsequent increase in fat mass under isoenergetic conditions [50].

Recently, we published an overview of the complex and bidirectional interlinks between the gut microbiome and the brain [29]. We have also published a discussion of the bidirectional interlinks of the gut microbiome with the liver [7] and the peripheral immune-inflammatory systems [19,29]. In this subsection, we provide a key summary of the evidence that links the gut microbiome (and/or their metabolic by-products) with central hypothalamic control of appetite and metabolism, implicating entero-endocrine, autonomic, and neuro-humeral pathways, summarized in Table 1 [40,51].

### 5.1. Entero-Endocrine Pathway

The gut microbiome impacts the hypothalamic control of appetite and metabolism via effects on the modulation of hormonal signals from entero-endocrine cells within the gut wall [29]. Some of these effects are indirect and stem from metabolic by-products of the gut microbiome (released during their utilisation of food for energy). These metabolic by-products include tryptophan metabolites, secondary bile acids, and SCFAs [40,52]. SCFAs are precursors of odd-chain fatty acids (bioactive compounds), and as such, these lipid species can be used as biomarkers for fibre-rich diets [53,54,55]. SCFAs are produced by caecal anaerobic microbes such as *Enterococcus* in the process of fermentation of dietary fibre [29]. SCFAs stimulate G protein-coupled receptors (GPRs), including GPR41 and GPR43 expressed on entero-endocrine cells [56]. In a rodent-based study that compared wild-type with GPR41 knockout mice, SCFAs stimulated GPR41 receptors, resulting in enhanced secretion of peptide YY (PYY, a potent appetite-suppressant gut-derived incretin-like hormone) only in the wild-type mice [57].

To complement the effects on GPR41, SCFAs also stimulate GPR43, resulting in the release of glucagon-like peptide-1 (GLP-1) that induces satiety and thereby supports hypothalamic appetite control [58]. Finally, SCFAs may translocate across the gut wall into the systemic circulation, and from there cross the blood–brain barrier into the brain parenchyma to influence the hypothalamic control of appetite and metabolism [59,60]. Through indirect effects on the entero-endocrine release of key incretin hormones like GLP1, and through direct central effects, SCFAs may meaningfully impact the hypothalamic control of appetite and metabolism and thereby represent a biological contributor to BMI. To corroborate this assertion, in a study on obese women undergoing bariatric surgery, colonic levels of *Enterococcus* were associated with appetitive inhibition [61]. Furthermore, in a randomized, controlled crossover study on overweight human adults, compared with inulin ingestion (control group), the ingestion of propionate (a common SCFA released by human gut microbiota) resulted in early postprandial release of the intestinal hormone PYY, and the incretin hormone GLP-1 from colonic entero-endocrine cells and reduced caloric intake. Metabolic benefits of regular propionate ingestion occurred over 24 weeks, with reduced hepatic lipid content and intra-abdominal adipose tissue volume, preserved insulin sensitivity, and significant weight loss [62]. However, the role of PYY in appetite regulation and its association with human body weight is controversial [63,64]. Furthermore, inulin (as a prebiotic) is fermentable and may also elicit metabolic effects of SCFAs over a longer time period [65]. Therefore, inulin is not an ideal control comparator for propionate ingestion.

Despite the suggested metabolic benefits of SCFAs demonstrated in some rodent- and human-based studies, these remain contentious. In a human-based study, compared with placebo, a 3-week treatment with oral prebiotic acetylated and butyrylated high-amylose maize starch in participants with essential hypertension resulted in an increase in serum levels of acetate and butyrate, and a clinically relevant reduction in 24-hour systolic blood pressure [66]. Conversely, in a separate human-based study, a 4-week treatment with oral butyrate resulted in a significant increase in both daytime systolic and diastolic blood pressure compared with the placebo group, in participants with hypertension [67]. Furthermore, SCFAs rely upon the fermentation of soluble dietary fibre [29]. In the short term, the ingestion of soluble (and highly fermentable) fibre improves insulin sensitivity and has some short-term glycometabolic benefits with anti-inflammatory effects [68]. However, the longer-term metabolic effects of dietary soluble fibre are less clear [68]. Conversely, there is much evidence to support the metabolic benefits of ingested insoluble cereal fibres (little-fermentable oat extracts, wheat, and whole grain products [69]), including improved insulin sensitivity and reduced risk for the development of T2D [43,70,71,72,73,74]. However, the beneficial long-term effects of insoluble cereal fibres are only observed in reported observational studies. Furthermore, interventional studies are sparse, particularly those that compare the metabolic effects of ingested insoluble with soluble fibre, such as one of the longest-duration rodent-based studies reported on to date, a 45-week study of an isoenergetic diet that compared the metabolic effects of additional dietary soluble fibre versus insoluble cereal fibre, with a matched high-fat diet in obesity-prone mice [75]. In this study, compared with the insoluble fibre mice, those fed soluble fibre had a greater production of SCFAs via colonic fermentation, a significant reduction in energy loss via the faeces, a significant increase in body weight, and elevated markers of insulin resistance. Furthermore, gene expression analysis in white adipose tissue revealed a significant increase in the levels of the fatty acid target G-protein coupled receptor-40 in the mice that were fed soluble fibre. Conversely, liver gene expression in the insoluble fibre group showed a pattern consistent with increased fatty acid oxidation. Therefore, there were clear differences in the metabolic effects (including body weight and measures of insulin resistance) of soluble versus insoluble dietary fibre added to a high-fat Western-style diet in obesity-prone mice.

Therefore, although dietary intake of soluble fibre results in significantly increased production of SCFAs that may have beneficial short-term metabolic benefits, these may be outweighed, at least in rodents, by adverse metabolic outcomes over the longer term. Arguably, in the rodent-based study outlined above [75], there was a relative overdose of dietary fibre and this may have explained the rather atypical weight gain for the mice assigned to the soluble fibre group mice (the dosage of fibre was 100 g per kg of food ingested, and in humans this would represent around 200 g of soluble dietary fibre per day). Interestingly, a somewhat similar study from China assessed the metabolic effects of dietary ingestion of 40 g of resistant starch in humans with low baseline dietary fibre intake, revealing body weight loss under these conditions [76]. Finally, in a 67-week duration rodent-based study, adolescent rats fed guar gum or guar by-product diets gained less weight than those fed cellulose, and only those fed guar gum had improved carbohydrate tolerance [77].

In addition to the stimulated release of GLP1 (via SCFAs), the gut microbiota may also influence the release of another incretin hormone, glucose-dependent insulinotropic polypeptide (GIP) [78]. In recent times, there has been much interest in the pharmacotherapeutic potential of GIP (in combination with GLP1 agonism) in the context of obesity and T2D [79]. GIP may exert some of its metabolic effects indirectly via the suppression of ghrelin release [80]. However, unlike GLP1, insights from rodent-based models reveal that the direct effects of GIP may be less desirable metabolically, including the mediation of energy intake and markers of insulin resistance in response to carbohydrate ingestion [81]. Indeed, such metabolic benefits of suppressed GIP release provide a rationale for the ‘endo-barrier’ as a weight-loss intervention (through creating a physical barrier between nutrients and the duodenal wall) [82]. In a rodent-based model, C57BL/6 mice and GIP-receptor knockout mice were exposed to ovariectomy (simulating menopause) or sham operation and observed for 26 weeks for changes in metabolic regulation and body weight [83]. The ovariectomized wild-type mice gained weight with increased fat mass and insulin resistance as expected, through effects of reduced oestrogen on the regulation of body weight, mediated in part by GIP signaling [83]. Conversely, the ovariectomized GIP-receptor knockout mice were protected from these adverse metabolic outcomes with a significant reduction in food intake, stemming from changes in the neurobiological regulation of the hypothalamic appetite centre [83]. This may provide one explanation for the weight neutrality of dipeptidyl peptidase-4 (DPP4) inhibitors that inhibit the breakdown of endogenous GLP1 and GIP.

### 5.2. Autonomic and Neuro-Humeral Pathways

The gut microbiota may influence the levels of important neurotransmitters and receptivity of neuroreceptors within the brain that regulate key physiological and psychological processes and emotional behaviours (including depression and anxiety) [40,84,85]. Gamma-amino butyric acid (GABA) is one such major inhibitory neurotransmitter [33,40]. Bidirectional signaling via the vagus nerve connects the gut microbiota with the brain. Such signaling may impact the hypothalamic control of appetite and metabolism indirectly via effects on mood and the functioning of the hypothalamic–pituitary–adrenal (HPA) axis. In a rodent-based study, chronic ingestion of *Lactobacillus rhamnosus* (JB-1), compared with control-fed mice, resulted in regionally dependent changes in the expression of GABA receptors within the brain that in turn associated with reduced depression- and anxiety-related behaviour [40]. The absence of such effects in vagotomised mice supports an important role for the vagus nerve in the mediation of signals between the gut microbiota and the brain [40]. The vagus nerve also links the gut microbiota with the liver, which in turn communicates with the hypothalamus to control appetite, feeding behaviour, and metabolism through myriad ways that include the release of hepatokines [86]. In addition to GABA, the gut microbiota influence the expression of other neuroreceptors within the brain, such as those for N-methyl-D-aspartate (NMDA, mediating effects of the excitatory neurotransmitter glutamate) [85], serotonin receptor 1A [84], and tryptophan (as a precursor to the neurotransmitter serotonin) [87]. Given the complexities of the neurobiology of central control of appetite and metabolism, including the impact of mood and emotions [88,89], it is likely that such neuro-humeral changes in response to the gut microbiota mediate changes in body weight and BMI.

## 6. Heritability of the Gut Microbiome

The gut microbiome can only contribute towards the heritability of BMI if the gut microbiome is itself heritable. In this section, we explore the available evidence to support the heritability of the gut microbiome and its transmissibility between generations. This discussion will include a role for vaginal delivery in the establishment of the gut microbiota in the newborn, and evidence that breast milk contains maternal gut microbiota as a means of seeding the gut microbiota of early infancy. We also consider the shared food environment of offspring and parents that influences the infant gut microbiota. Although such a shared environment would not strictly constitute a ‘heritable’ mechanism, it may nonetheless help to explain some of the familial concordance of BMI.

### 6.1. Vaginal Delivery in the Establishment of the Gut Microbiome of the Newborn

There has been much interest in the impact of the mode of delivery on the initial establishment of the gut microbiota and future health implications. Caesarean section (C-section) eliminates contact of the newborn with maternal microbes during parturition, and in this scenario, the newborn gut microbiota are derived from the environment (including possible pathogens), with potentially negative implications for long-term metabolic and immunological functioning [90]. Intrapartum antibiotic exposure can also diminish the transfer of maternal gut microbiota to the offspring during vaginal delivery [90]. (Following C-section, there may still be a role for breastfeeding in the establishment and nurture of the newborn gut microbiota). Several large cohort studies and meta-analyses have revealed an association between C-section birth and long-term health problems, including obesity and chronic inflammatory and immune diseases that may extend into adulthood [90,91]. Although confounding factors exist, given the important effects of the gut microbiota on physiological and immunological development, these data are consistent with a lack of protection from maternally derived gut microbiota (during vaginal delivery) for offspring born through C-section [90].

In contrast to C-section, during vaginal delivery, there is oral exposure of the newborn to the maternal microbiome. In addition to vaginal microbes and skin commensals, this also includes oral exposure to the maternal gut microbiome [92]. In a study that used quantitative PCR, there were significant differences in the composition of the gut microbiota in young adults based on their mode of delivery (vaginal vs. C-section) [90,93]. In a systematic review that explored the effect of the mode of delivery on the gut microbiota of newborn infants, it was shown that compared with C-section, vaginal delivery resulted in a greater overall diversity and colonisation pattern of the infant gut microbiota (including significantly more *Bifidobacterium* and *Bacteroides* genera) during the first 3 months of life [94]. *Bifidobacteria* are important for early gut health and have widespread immunological effects that facilitate the normal development of the immune system [90]. Low gut microbial abundance of *Bifidobacteria* (especially *B. longum*) early in life is associated with an increased risk of allergic diseases later in childhood [95] and is predictive of adiposity later in life [90,96]. During vaginal delivery, maternally derived microbes colonise the infant gut permanently, whereas non-maternal gut microbes are transient typically [90,97]. The permanence of maternally derived microbes within the gut suggests compatibility between maternal and infant gut microbes that may be mediated genetically through immune factors and gut mucus composition [90].

The published data outlined here support a role for vaginal delivery in the early establishment (and permanence) of the infant gut microbiota with future health implications. Furthermore, vaginal delivery enables the transmission of maternal gut microbiota to her offspring, therefore supporting the heritability of gut microbiota in vaginally delivered infants.

### 6.2. Breastfeeding as a Means of Seeding the Gut Microbiome of Early Infancy

Although traditionally considered as sterile, over the past two decades, our understanding of human breast milk (HBM) constituents has been transformed by the identification of commensal bacteria [98], including lactic acid bacteria (LAB) genetically distinct from the LAB isolated from the maternal breast skin and mammary areola [98,99]. Within healthy HBM, *Staphylococcus* and *Streptococcus* appear as universally predominant genera [98,100]. However, there is a rich diversity of the HBM microbiome, including *Firmicutes* and *Proteobacteria* [98,101]. It is now widely accepted that HBM contains an abundance of diverse microorganisms [98,101], and that these play a key role in the colonisation of the infant gut and the establishment of the infant gut microbiome [98,102]. As such, the transmission of maternal microbiota to the offspring via HBM may represent a heritable biological trait. However, this would depend on the translocation of maternal gut microbiota via an ‘entero-mammary pathway’ (EMP).

The origin of the HBM microbiome is incompletely understood [98]. In addition to the EMP, another theory is that the HBM microbiome develops secondary to contamination from maternal surface skin and the infant’s oral cavity [98]. Early studies seemed to favour the latter hypothesis, based on similarities between the HBM microbiome and skin micro-organisms (such as *Staphylococcus* and *Corynebacterium*) [98,103,104]. However, there are marked differences between the microbiome of the skin and HBM [98,105,106]. Furthermore, anaerobic bacteria such as *Bacteroides*, *Clostridium*, *Bifidobacterium*, and *Parabacteroides* are shared between HBM and infant faeces but are absent from adult skin [98,99,107]. Therefore, it seems unlikely that the HBM microbiome originates from the maternal breast skin and mammary areola. An EMP would implicate the translocation of maternal gut microbiota across the gut wall, followed by endogenous transfer to the mammary glands [98]. Consistent with this hypothesis is the presence of an entero-mammary circulation of IgA-producing cells [98,108], and the observation that dendritic cells can penetrate the gut wall and open tight junctions that then allow the translocation of the gut microbiota across the gut wall [98,109]. Furthermore, some microbiota species such as *Bifidobacterium* and *Lactobacillus* are shared between the maternal HBM and the infant feces [98,99,104,110]. There is also evidence for the presence of extracellular vesicles (EVs) derived from bacteria (especially *Bifidobacterium* and *Lactobacillus*) within HBM [98,111,112] that likely facilitate the microbiota colonisation of the infant gut and may act as receptors for bioactive molecules in the host cell [98,111,113]. These observations are all consistent with the existence of an EMP that enables the transmission of maternal gut microbiota to her offspring, thereby supporting the heritability of gut microbiota in breast-fed infants. Future research should explore the exact mechanisms that underlie the EMP.

### 6.3. Shared Food Environment of Offspring and Parents and the Gut Microbiome

Although traditionally considered a non-communicable disease, there is an argument for obesity being a communicable disease, mediated, for example, by dietary habits that are shared between families and indeed whole communities [114]. Heritability refers to the transmission of biological traits between generations and as such does not include environmental factors. However, it is known that environmental and lifestyle factors, particularly diet, have an impact on the composition of the gut microbiome. Therefore, shared food environments between parents and offspring may account for some familial traits of the gut microbiota [115]. Indeed, in a three-generational cohort study based on a Dutch population, it was shown that around 48.6% of taxa within the gut microbiota were significantly explained by cohabitation (with only 6.6% of taxa being heritable) [116]. We reviewed the impact of diet on the gut microbiome in detail previously [29]. Here, we provide a summary.

In the industrialized era, Western diets have changed radically, with diminished dietary fibre (from unprocessed plant-based foods), coupled with an abundance of ultra-highly processed foods, laden with additional fats and carbohydrates [46]. Evidence from rodent-based studies reveals that even within a single day, the gut microbiota alters in response to changes in dietary macronutrient intake [117]. Human-based studies show variable timeframes for diet to impact the composition of the gut microbiota, including the absence of any significant changes in the gut microbiota in response to dietary changes [118], limited diet-related changes to the composition of the gut microbiota occurring over weeks or months [119,120,121], and changes to the gut microbiota manifesting over the short term in response to changes in the diet [122]. In a well-phenotyped study using metagenomic sequencing, the ingestion of healthy and diverse plant-based foods was a key factor that promoted a significant association between gut microbes and specific nutrients and food groups [123]. Furthermore, the composition of the gut microbiota was predictive of multiple cardio-metabolic blood markers [123]. There is clear evidence from the literature that dietary fibre (including both fermentable (mostly soluble) and little-fermentable (often insoluble) fibres), derived from plant-based foods, plays a key role in the establishment and nurture of a healthy gut microbiome and the promotion of health and wellbeing [46]. Soluble fibre is associated with *Bacteroides* species and butyrate-producing bacteria (*Clostridium leptum* and *Eubacterium rectale*) [124,125]. Furthermore, the ingestion of complex carbohydrates promotes the growth of *Bifidobacteria* species (*Bifidobacterium longum, Bifidobacterium breve*, and *Bacteroides thetaiotaomicron*), each of which is favourable for health [126].

One caveat to a shared food environment between parents and offspring is that offspring (particularly older children and adolescents) often have diets and food preferences that differ from their parents [127]. However, cohabiting families do share their food and meal environments, and children model their eating behaviours, food choices, and taste preferences from their parents [128]. Therefore, even with some generational differences in food preferences, cohabitation results in at least a partial confluence of gut microbiota between parents and their offspring.

## 7. Conclusions and Future Directions

To summarize, common polygenic obesity, unlike monogenic conditions, has a complex and incompletely understood underlying pathogenesis that reflects an elaborate interplay between our genetic architecture and modern-day obesogenic environment. BMI is a heritable trait, but GWAS studies have only identified a small proportion of the overall heritability of BMI. Given that heritability incorporates any inherited biological trait, it is important to search beyond the human genome to identify some of this missing heritability. It seems likely that the gut microbiome contributes to the heritability of BMI [129,130] given its impact on the control of hypothalamic appetite and metabolic control through entero-endocrine, autonomic and neuro-humeral effects, and the maternal–offspring intergenerational transmissibility of the gut microbiome, at least in the context of vaginal delivery and/or breastfeeding.

There are limitations to our outlined hypothesis. Our evidence for the impact of gut microbiota on the hypothalamic control of appetite and metabolism stems primarily from rodent-based studies, with a relative lack of human-based studies. It is important not to over-extrapolate the implications of data between species, and it remains possible that the gut microbiota play less of a role in appetite and metabolic control in humans than in rodents, although we need to await further data from human-based studies to make further conclusions on this point. Furthermore, there are confounders that are implicit in the data outlined, including, for example, associations between mode of delivery and future risk of chronic illness and BMI. Our hypothesis is based on observational data, with obvious difficulties, hurdles, and challenges for any future randomized controlled trials in this field.

The dynamicity and modifiability of the gut microbiome present an opportunity for the future prevention of weight gain and management of obesity. Numerous lifestyle factors including physical activity, sleep, and stress influence the gut microbiome in important ways [29]. As outlined, dietary factors (particularly the ingestion of diverse and unprocessed plant-based foods) are also important for the nurturing of a healthy gut microbiome [29]. There is a need for further human-based studies to explore the potential metabolic and weight loss benefits of FT or other therapeutic strategies to transform the gut microbiota, thereby extending the indications for FT beyond the management of intractable colonic colonisation with *Clostridium difficile* [29,131]. Furthermore, given the intergenerational transmissibility of the maternal gut microbiota to offspring during vaginal delivery and breastfeeding, there is a case for optimisation of the maternal gut microbiome pre-conception, analogous to our current approach to optimize maternal body weight and glycaemic control pre-conception.

To conclude, it is useful to provide learning points from our review. Our gut microbes influence our health and wellbeing through myriad mechanisms. We should all strive to nurture our gut microbiome through healthy lifestyles and dietary intake of diverse, unprocessed plant-based foods. This should include the ingestion of fermented plant-based foods such as kimchi, sauerkraut, and natto, and dairy-based foods such as kefir [132]. Other aspects of our gut microbiome manifest a degree of permanence, particularly our maternally derived microbiota from infancy. Given the heritability of our gut microbiome, at least in the context of vaginal delivery and/or breastfed infants, it is important for reproductive-age women to strive for optimisation of their gut microbiome through healthy lifestyle and dietary strategies (as outlined above) throughout pre-conception and extending into antenatal and postnatal periods. Finally, it is important that insights from our review are disseminated broadly throughout society to address the widespread misconceptions and myths regarding the development of weight gain and obesity that are often promulgated and stigmatized by the mass media. Improved public understanding of the heritability of BMI, including the role of genetic factors and other biological and heritable traits such as our gut microbiome, would help to address societal myths and misconceptions regarding the origins of obesity. Such insights would also help to encourage and promote the widespread adoption of healthy lifestyles and diets. Finally, we live in a cruel and unfair society in which human myths abound (to a greater extent than what most of us appreciate), in which weight gain and obesity are often incorrectly attributed to human vices, redolent of a medieval mindset in which plague and other poorly understood diseases were attributed to the wrath of God as a punishment for human misdemeanors (with associated stigmatisation of such diseases). Improved public understanding of obesity pathogenesis, including an appreciation of our genetic misalignment to our modern-day obesogenic environment, and that our BMI is largely biologically inherited through both our genetics and our gut microbiome, would help to foster cultural change apropos of societal attitudes towards people living with obesity. A cultural change in which there is vanquishment of obesity-related stigma and abuse, and a new dawn of understanding, support, and compassion.

## Figures and Tables

**Figure 1 nutrients-17-01713-f001:**
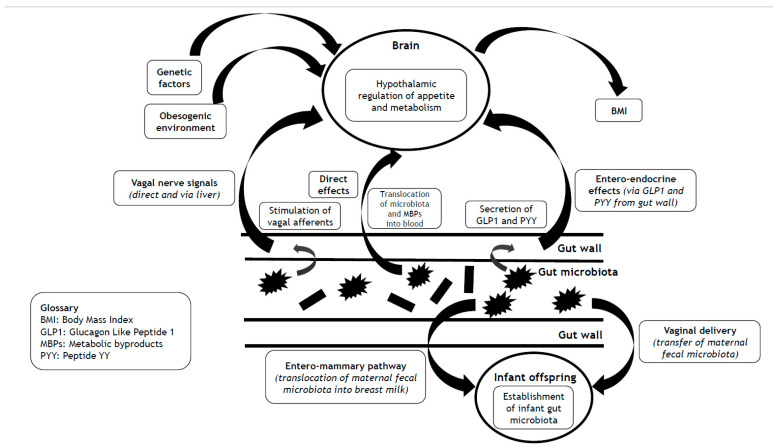
Impact of the gut microbiota on BMI via hypothalamic control of appetite and metabolism, and mechanisms that underlie the transgenerational effects of the gut microbiota on the heritability of BMI.

**Table 1 nutrients-17-01713-t001:** Summary of the mechanisms whereby the gut microbiota communicate with the brain.

Gut Microbiota Component	Derivation of Gut Microbiota Component	Communication Mode with Brain	Communication Pathway with Brain	Effects on Brain	Data Source
SCFAs	Metabolic by-product of gut microbiota	Enteroendocrine and direct (via vasculature)	SCFAs enhance the release of GLP1 and PYY from gut wall; cross the gut wall and BBB to influence brain function	Influence directly central control of appetite and metabolic pathways	Human and rodent
Extracellular vesicles	Released from gut microbiota	Direct (via vasculature)	Gut microbiota-derived extracellular vesicles cross the gut wall and BBB to influence brain function	Regulation of central gene expression and neuroinflammatory and neurodegenerative pathways	Human and rodent
Endotoxins (LPS)	Derived from the cell walls of Gram-negative gut microbiota	Direct (via vasculature)	LPS cross the gut wall and BBB to influence brain function	Regulation of both central and peripheral inflammatory processes and insulin sensitivity	Human and rodent
*Lactobacillus* *rhamnosus*	Gut microbiota	Vagal afferent signaling	Stimulation of vagal afferent termini within the gut wall	Neuro-humeral effects; expression of central GABA receptors and tryptophan and NMDA; effects on appetite and mood	Rodent

BBB: blood–brain barrier; GABA: gamma-amino butyric acid; GLP1: glucagon-like peptide-1; LPS: lipopolysaccharide; NMDA: N-methyl-D-aspartate; PYY: peptide YY; SCFAs: short-chain fatty acids.

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
