# Peer review of "The Gut Microbiome as a Key Determinant of the Heritability of Body Mass Index"

_nutrients, 2025, doi:10.3390/nu17101713_

Round 1
Reviewer 1 Report
Comments and Suggestions for Authors
The manuscript cannot be considered a review, but rather a kind of dissertation by the authors on the selected topic. It does not contain tables or figures to support the text, it mentions in generic manner the presence of genes, alleles, bacterial groups, etc. that are involved in the mechanisms that the authors intend to point out but they do not specify them.
The writing of most of the manuscript is in the first person, exposing past work or personal impressions. In many cases the information is not up-to-date, as in the expression “Recently, there has been much interest in the impact of mode of delivery on the initial establishment of the gut microbiota and future health implications.” Rciently...there has been research on this for more than a decade.
The format of the references in the text is not in accordance with MDPI recommendations. Neither is the list of references.
There are fragments of text written in a font and size different from the rest of the text, as in lines 282-287 or 385-392.
Of the total number of references included, 27 are self-citations by the authors themselves.
The search criteria mentioned do not allow any verification of the search performed. They only mention “The search terms were as follows: 'Body Mass Index; genetics; heritability; gut microbiota; appetite; vaginal delivery; breast feeding”, but without mentioning in which field they have searched (title, topic, all fields...etc), nor the cross searches performed. It is impossible for them to have searched the terms individually, as the number of articles found would be immense.
Both the abstract and the conclusions are excessively long and include digressions of the authors rather than conclusions of the work included in the text.
There are also large parts of the manuscript that are written without any documentary support, such as lines 51-64 or 77-96, which in many cases include conjectures and personal opinions.
In conclusion, as already mentioned, the manuscript is not a review, but a current opinion or perspective.
Author Response
Reviewer 1
Comment 1.1:
The manuscript cannot be considered a review, but rather a kind of dissertation by the authors on the selected topic. It does not contain tables or figures to support the text, it mentions in generic manner the presence of genes, alleles, bacterial groups, etc. that are involved in the mechanisms that the authors intend to point out but they do not specify them. In conclusion, as already mentioned, the manuscript is not a review, but a current opinion or perspective.
Response to comment 1.1:
Thank you for this comment. We agree with the reviewer that our review has a narrative style and contains the opinions and perspectives of the authors. However, for such a review, it is necessary to include the views and perspectives of the authors given that much of this field remains unknown and incompletely understood, such as the genetic heritability of BMI based on GWAS data, and the complexity of the gut microbiota, and its role in the regulation of appetite and body weight. We respectfully disagree with the reviewer regarding the classification of our paper as a review, given the high number of citations in our manuscript, and the inclusion of much discussion regarding recently published papers in the field. We would argue strongly therefore that our paper is a concise narrative review of the field, albeit one that also includes the views, perspectives, and learning points of the authors.
We included reference to specific gut bacterial species within the original manuscript. Due to the concise nature of our review, we were not able to provide a detailed exposition of the genetics of obesity, which has been provided in other recent reviews. This was not the purpose of our review, but rather to consider the role of the gut microbiota in the heritability of BMI.
In the original version of our manuscript, we did include a figure that complements and supports the text. We agree with the reviewer that an additional table would help with the readability of the text. Accordingly, we have included a table that summarizes much of section 5 in the revised version of our manuscript.
Comment 1.2:
The writing of most of the manuscript is in the first person, exposing past work or personal impressions. In many cases the information is not up-to-date, as in the expression “Recently, there has been much interest in the impact of mode of delivery on the initial establishment of the gut microbiota and future health implications.” Recently...there has been research on this for more than a decade.
Response to comment 1.2:
Thank you for this comment. We agree with the reviewer. In response, we have deleted the word ‘recently’ from the relevant sections of the revised version of our manuscript.
Comment 1.3:
The format of the references in the text is not in accordance with MDPI recommendations. Neither is the list of references.
Response to comment 1.3:
Thank you for this comment. We believe that the format and list of references in the revised version of our manuscript are in accordance with MDPI recommendations.
Comment 1.4:
There are fragments of text written in a font and size different from the rest of the text, as in lines 282-287 or 385-392.
Response to comment 1.4:
Thank you for this comment. We apologise for this and assume that the font of these lines of text was changed for some reason during the transfer of our manuscript to the Nutrients template. We have corrected and unified the font throughout the revised manuscript.
Comment 1.5:
Of the total number of references included, 27 are self-citations by the authors themselves.
Response to comment 1.5:
Thank you for this comment. We agree with the reviewer that we had provided a large number of self-citations, and have now reduced the percentage of self-citations in the revised version of the manuscript. Please see also our response to the Editor’s comment.
Comment 1.6:
The search criteria mentioned do not allow any verification of the search performed. They only mention “The search terms were as follows: 'Body Mass Index; genetics; heritability; gut microbiota; appetite; vaginal delivery; breast feeding”, but without mentioning in which field they have searched (title, topic, all fields...etc), nor the cross searches performed. It is impossible for them to have searched the terms individually, as the number of articles found would be immense.
Response to comment 1.6:
Thank you for this comment. We understand the concern raised by the reviewer. To clarify, the search terms refer only to the title of published articles. We agree with the reviewer that the number of articles in this field is immense. Therefore, it was necessary for us to select the most recent and impactful published articles within the field for our narrative review. To address these points, we have included some additional text to the methodology section of the revised version of our manuscript.
Comment 1.7:
Both the abstract and the conclusions are excessively long and include digressions of the authors rather than conclusions of the work included in the text.
Response to comment 1.7:
Thank you for this comment. We agree with the reviewer. Accordingly, we have deleted some of the redundant text from both the abstract and conclusion, thereby truncating these sections in the revised version of our manuscript.
Comment 1.8:
There are also large parts of the manuscript that are written without any documentary support, such as lines 51-64 or 77-96, which in many cases include conjectures and personal opinions.
Response to comment 1.8:
Thank you for this comment. We agree with the reviewer. In response, we have included some additional references in the text of the two paragraphs referred to in the introduction of the revised version of our manuscript. In this way, we have included documentary and published support of many of the points made in these sections. We would argue that with this type of narrative review, on a topic that is incompletely understood, it is important to include some personal views and conjectures from the authors.
Reviewer 2 Report
Comments and Suggestions for Authors
All comments were provided in the attached document.

Author Response
Reviewer 2
Comment 2.1:
A brief summary- The manuscript is a succinct summary of the knowledge on the transferability of obesity through the intestinal microbiome. This is about a 20-year-old concept that requires further research, as outlined by the authors. The manuscript includes background information explaining the alleged link between the heritability of obesity and the microbiome, and dives into the pathways through which the microbiome may be inherited and through which it may impact the BMI.
Response to comment 2.1:
Thank you for this comment. We agree with the reviewer.
Comment 2.2:
General concept comments—The concept is well known in the literature, and many reviews have been published on this and related topics. Nevertheless, the continuous trend of studying the relationship between obesity and the human gut microbiome indicates that there are many unknowns that require further exploration. Therefore, reviews updating the state of knowledge are important. Personally, I would rather see a systematic review with or without meta-analysis that would be based on recent (e.g. last 5 years) studies, but reading a well-written narrative review, such as the present manuscript, is also of interest to myself and possibly many other readers who look for a succinct summary on the subject.
Response to comment 2.2:
Thank you for this comment. We agree with the reviewer.
Comment 2.3:
The quality of English throughout the whole text is very good. The text is well written, organised and conveys the message well. There are not many publications with such a high-reading quality! Nevertheless, the enjoyment of reading is an important factor that attracts citations. Therefore, this paper, despite being a narrative review, may be impactful. Overall, the content of the manuscript is interesting, and I would recommend only some minor revisions outlined in the specific comments.
Response to comment 2.3:
Thank you for this comment.
Comment 2.4:
Title- is brief, but I would recommend rephrasing it to include the word heritability, which is one of the features within this review.
Response to comment 2.4:
Thank you for this comment. We agree with the reviewer. In response, we have modified the title in the revised manuscript as: ‘The Gut Microbiome as a key determinant of the Heritability of Body Mass Index’.
Comment 2.5:
Abstract- The abstract is quite long. Not sure if it meets Nutrient’s criteria. But it is well written. The authors contained in it an overview of the review’s content and finalised it with a broad range of recommendations. This section reads very well and is structured appropriately. Keywords- are appropriate.
Response to comment 2.5:
Thank you for this comment. We agree with the reviewer. We confirm that the abstract meets the criteria for Nutrients.
Comment 2.6:
Introduction A very well-written section. I would recommend adding a sentence or two on the drawbacks (side effects) of pharmacotherapies in use for obesity treatment, and the importance of introducing systemic solutions that could use a microbiome-based targeted nutrition approach.
Response to comment 2.6:
Thank you for this comment. We agree with the reviewer. In response, we have included some additional text and reference to the introduction section of the revised version of our manuscript.
Comment 2.7:
Methodology This section is of appropriate depth for a narrative review, although when it comes to well-known topics, such as the link between BMI and microbiome, I would rather see a synthetic summary of recent literature than a search of publications from all times.
Response to comment 2.7:
Thank you for this comment. Please kindly see our response to comment 1.6.
Comment 2.8:
The Heritability of BMI If one did not read the abstract before reading the article, this section would appear as disjointed from the main topic. Perhaps authors could include the word heritability in the article’s title? Besides this, the section provides a good introduction to why the microbiome may be a missing link that could explain the heritability of BMI.
Response to comment 2.8:
Thank you for this comment. We agree with the reviewer. Regarding the modification of the title in the revised version of our manuscript, please see our response to comment 2.4.
Comment 2.9:
The Gut Microbiome and the Metagenome Only the second, shorter paragraph refers to the section topic. Authors could consider merging this section with another one. It aims to introduce the concept of the metagenome. Perhaps this could be done in the introduction?
Response to comment 2.9:
Thank you for this comment. We agree with the reviewer that the title was unclear for section 4 in the original version of our manuscript. However, we do feel that a separate section on the gut microbiome is important and helps to break up the text to make it more meaningful for the reader and helps to introduce this topic for the rest of the review. In response, we have adjusted the title for section 4 to ‘the gut microbiome’ and have deleted some of the redundant text from this section to make it shorted in the revised version of our manuscript.
Comment 2.10:
Interlinks Between the Gut Microbiome and Central Appetitive and Metabolic Control This section is very complex. It is divided to appropriate subsections, but a figure that would summarise and guide through its content at the beginning could prepare the reader. In the first paragraph, the authors included information about the communication of the microbiota with the human microorganism. There has been a lot of research on the implied role of extracellular vesicles produced by the microbiota in the gut-brain axis. This phenomenon deserves a sentence or two in such a review.
Response to comment 2.10:
Thank you for this comment. Regarding the additional figure, please see our response to reviewer 1. We have included a table that summarises much of section 5 in the revised version of our manuscript. We agree with the reviewer regarding the inclusion of discussion on the role of extracellular vesicles produced by the microbiota on the gut-brain axis. Accordingly, we have included some additional text and reference in section 5 of the revised version of our manuscript.
Comment 2.11:
Lines 282-287; 365-368; 385-391- the font here is different from the rest of the text Lines 295-314- this paragraph mentions rodent-based studies; there are many human-based studies where changes in the microbiome were measured following the weight loss, there are also studies on the effect of faecal transplants from lean to obese humans, Therefore there is not much rationale behind introducing well-known rodent-based studies. The authors should consider changing their choice of literature here.
Response to comment 2.11:
Thank you for this comment. Regarding the formatting issue, please see our response to comment 1.4. Regarding the paragraph referred to in section 5, we agree with the reviewer that there should be more of a focus on human- rather than rodent-based studies. Accordingly, we have included some additional text and references regarding human-based studies and re-structured this paragraph to focus more on the human-based literature in the revised version of our manuscript.
Comment 2.12:
Lines 365-385- reference 63 appears in each sentence. Citing once, at the beginning, would suffice. This study is interesting and quite important here; perhaps the authors could consider summarising its findings in a figure?
Response to comment 2.12:
Thank you for this comment. We agree with the reviewer. In response, we have only included the suggested reference once at the beginning of this section in the revised version of our manuscript. Regarding the additional figure, please see our response to reviewer 1. We have included a table that summarises much of section 5 in the revised version of our manuscript
Comment 2.13:
The adverse effects of SCFAs are also known from human studies. This is a study where individuals with hypertension responded with an increase of blood pressure to butyrate supplementation (https://pubmed.ncbi.nlm.nih.gov/39034917/) and another one where fibre resulted in an increase of SCFA production by microbes, and an inverse effect was observed (https://www.nature.com/articles/s44161-022-00197-4). The authors could consider adding these reports to the list of references and highlighting the ambiguity of the evidence.
Response to comment 2.13:
Thank you for this comment. We agree with the reviewer. Accordingly, we have included some additional text and the suggested references to section 5.1 of the revised version of our manuscript.
Comment 2.14:
Line 420- Lactobacillus should start with a capital letter, and it would be appropriate to mention the specific designation of this strain, as well as the reference to the whole sentence. Subsections 5.2 and 5.3 both refer to GABA, and perhaps could be merged, given that both subsections are very short.
Response to comment 2.14:
Thank you for this comment. We agree with the reviewer regarding the correct nomenclature of Lactobacillus and inclusion of a reference. We also agree with the reviewer that subsections 5.2 and 5.3 could be merged into one sub-section. Accordingly, we have modified the text and conflated these two sub-sections into one in the revised version of our manuscript. We have also deleted some text in the original sub-section 5.3 that was a repetition of text in the original sub-section 5.2 in the revised version of our manuscript.
Comment 2.15:
Heritability of the Gut Microbiome The whole section summarises the most important elements of current knowledge well (especially subsections 6.2 and 6.3). I would only recommend the inclusion of a summary figure at the beginning of the section.
Response to comment 2.15:
Thank you for this comment. Regarding the additional figure, please see our response to reviewer 1. We have included a table that summarises much of section 5 in the revised version of our manuscript
Comment 2.16:
Lines 456-468- afterwards, maternal microbiota may be transferred via breast-feeding, authors should mention this when referring to c-section and be more specific as to what part of the maternal microbiota is not transferred due to the birth procedure.
Response to comment 2.16:
Thank you for this comment. We agree with the reviewer. In response, we have added some additional text to section 6.1 of the revised version of our manuscript, to improve the clarity of this sub-section.
Comment 2.17:
Section 6.3- There are interesting references indicating that obesity is a communicable disease, with one of the influencing factors- dietary habits that are similar in whole communities. I would recommend that authors refer to one such article at the beginning of this section.
Response to comment 2.17:
Thank you for this comment. We agree with the reviewer. In response, we have included some additional text and reference to the opening of the sub-section 6.3 in the revised version of our manuscript.
Comment 2.18:
Conclusions and Future Directions Read well, although it may be shortened by removing repetitions. In addition, authors could consider addressing following comments: Line 590- the sentence in this line summarises the content of the manuscript very well, authors could consider basing the manuscript title on this sentence.
Response to comment 2.18:
Thank you for this comment. Regarding the suggested title change based on line 590, please see our response to comment 2.4. In response, we have also deleted some of the text in the conclusions section of the revised version of our manuscript.
Comment 2.19:
Lines 606-608- consider moving to section 5 Line 620- kefir is not a plant-based food, consider rephrasing to connect better to the previous sentence
Response to comment 2.19:
Thank you for this comment. We agree with the reviewer. In response, we have moved this text from the conclusion to section 5 of the revised version of our manuscript. We have re-phrased the text relating to kefir in the conclusion section of the revised version of our manuscript.
Comment 2.20:
Figure 1 – Is quite complex, I would recommend dividing it into smaller pieces and presenting it as a summary of/ guide to each section. There is an online tool called Biorender, used by many scientists to produce high-quality colourful figures. High-quality figures are always a good addition to the review, helping to increase the interest of the readers and attract citations.
Response to comment 2.20:
Thank you for this comment. We understand the reviewer and agree that generally figures can be more impactful when shown in colour. However, with respect, we prefer to keep the figure in black and white for two main reasons. Firstly, figure 1 in the original version of our manuscript does not really require the addition of colour as it is mainly a diagrammatic representation of the pathways whereby the gut microbiota can impact on the control of appetite and metabolism, with arrows and text boxes. Arguably, such a representation is quite impactful when shown in black and white and there is not really a need for additional colour. Secondly, the production of colour images requires additional publishing charges, and as we have a restricted budget for this review, we are not able to cover these additional costs. Furthermore, we feel that the figure would lose some of its meaning if it was fragmented and sub-divided with components placed in each section. Our aim was to summarise all the mechanisms discussed in the review in one figure. Therefore, with respect, we would prefer to use figure 1 as set out in the original version of our manuscript.
Comment 2.21:
References- The list contains an appropriate number of references for a narrative review (122). All of the references presented in the current version of the manuscript are justified, although some rodent studies could be replaced by more recent human studies (as outlined in comments to relevant subsections.
Response to comment 2.21:
Thank you for this comment. We agree with the reviewer. Please see our response to comment 2.11.
Reviewer 3 Report
Comments and Suggestions for Authors
The authors present the complex associations of the gut microbiome with obesity.
They conducted an in-depth analysis of the influence of genetic and environmental factors on obesity, which became the starting point for considering the microbiome and metagenome. The entire text indicates the authors' extensive knowledge of the pathogenesis of obesity and an excellent understanding of the cited publications. However, the most valuable aspect of the paper is the authors' thoughts and comments on the discussed topics.
In addition, one must emphasise the dawn writing style; the article reads like a thrilling short story.
Reviewing such an article is a pleasure.
Author Response
Reviewer 3
Comment 3.1:
The authors present the complex associations of the gut microbiome with obesity.
They conducted an in-depth analysis of the influence of genetic and environmental factors on obesity, which became the starting point for considering the microbiome and metagenome. The entire text indicates the authors' extensive knowledge of the pathogenesis of obesity and an excellent understanding of the cited publications. However, the most valuable aspect of the paper is the authors' thoughts and comments on the discussed topics. In addition, one must emphasise the dawn writing style; the article reads like a thrilling short story. Reviewing such an article is a pleasure.
Response to comment 3.1:
We would like to thank the reviewer for their kind words, this is much appreciated.
Round 2
Reviewer 1 Report
Comments and Suggestions for Authors
The manuscript was strongly imporved and now it is aceptable for its publication. There are a misatke in space in reference 42, text was avoided.
Author Response
We thank the reviewer for this comment.
Please kindly note, there is no error in list of references. For ref 43, the page numbers are 439-442. Due to referencing style, it is listed as 439-42, and it is unfortunate that the end page number ‘42’ falls on next line, so has appearance of an additional ref 42, but it is actually page number for ref 43.